# High-pressure synthesis of seven lanthanum hydrides with a significant variability of hydrogen content

Dominique Laniel [1,2] ✉, Florian Trybel[3], Bjoern Winkler[4], Florian Knoop [3], Timofey Fedotenko[1], Saiana Khandarkhaeva[1], Alena Aslandukova[5], Thomas Meier[6], Stella Chariton[7], Konstantin Glazyrin [8], Victor Milman [9], Vitali Prakapenka [7], Igor A. Abrikosov [3], Leonid Dubrovinsky [5] & Natalia Dubrovinskaia [1,3]

The lanthanum-hydrogen system has attracted significant attention following the report of superconductivity in $LaH_{10}$ at near-ambient temperatures and high pressures. Phases other than $LaH_{10}$ are suspected to be synthesized based on both powder X-ray diffraction and resistivity data, although they have not yet been identified. Here, we present the results of our single-crystal X-ray diffraction studies on this system, supported by density functional theory calculations, which reveal an unexpected chemical and structural diversity of lanthanum hydrides synthesized in the range of 50 to 180 GPa. Seven lanthanum hydrides were produced, $LaH_3$, $LaH_{-4}$, $LaH_{4+\delta}$, $La_4H_{23}$, $LaH_{6+\delta}$, $LaH_{9+\delta}$, and $LaH_{10+\delta}$, and the atomic coordinates of lanthanum in their structures determined. The regularities in rare-earth element hydrides unveiled here provide clues to guide the search for other synthesizable hydrides and candidate high-temperature superconductors. The hydrogen content variability in lanthanum hydrides and the samples' phase heterogeneity underline the challenges related to assessing potentially superconducting phases and the nature of electronic transitions in high-pressure hydrides.

The report of superconductivity at the critical temperature ($T_c$) of 203 K at 150 GPa in the sulfur-hydrogen system[1] in 2015, followed by a gold rush in high-pressure sciences towards exceeding these temperatures, placed the dream of achieving room-temperature superconductivity within reach. Higher $T_c$ were claimed to be realized in La-H[2,3], Y-H[4,5], and in a carbonaceous sulfur hydride[6], the latter reaching a record value of 283 K at a pressure of 267 GPa. The compounds presumed to feature superconductivity in these systems could not be

recovered to ambient conditions, precluding further physical properties measurements, essential for a complete understanding of the mechanisms enabling ultra-high $T_c$.

At the same time, the reports on high $T_c$ in high-pressure hydrides have been heavily disputed[7–11]. The presence of phases other than the superconducting one has been considered as a potential explanation for the sudden drop of resistivity measured in some systems as they could enable a metallic conduction path to form upon temperature

[1]Material Physics and Technology at Extreme Conditions, Laboratory of Crystallography, University of Bayreuth, 95440 Bayreuth, Germany. [2]Centre for Science at Extreme Conditions and School of Physics and Astronomy, University of Edinburgh, EH9 3FD Edinburgh, UK. [3]Department of Physics, Chemistry and Biology (IFM), Linköping University, SE-581 83 Linköping, Sweden. [4]Institut für Geowissenschaften, Abteilung Kristallographie, Johann Wolfgang-Goethe-Universität Frankfurt, Altenhöferallee 1, D-60438 Frankfurt am Main, Germany. [5]Bayerisches Geoinstitut, University of Bayreuth, 95440 Bayreuth, Germany. [6]Center for High Pressure Science & Technology Advanced Research, Beijing, China. [7]Center for Advanced Radiation Sources, University of Chicago, Chicago, IL 60637, USA. [8]Deutsches Elektronen-Synchrotron, Notkestr. 85, 22607 Hamburg, Germany. [9]Dassault Systèmes BIOVIA, CB4 0WN Cambridge, UK. ✉e-mail: dominique.laniel@ed.ac.uk

decrease[7]. The lack of a homogeneous sample would not only affect resistivity measurements but also magnetic susceptibility[1,12,13] and nuclear magnetic resonance[14–16], serving as a strong *impetus* to accurately determine all phases present in these superconducting hydride systems. Single-crystal X-ray diffraction (SCXRD) coupled with ab initio calculations has already been demonstrated as a very effective tool to identify the formation of novel hydrides in laser-heated diamond anvil cells (DACs) experiments. For example, this methodology allowed the synthesis and the full characterization of two sulfur hydrides ($H_{6\pm x}S_5$ ($x \sim 0.4$) and $H_{2.85\pm y}S_2$ ($y \sim 0.35$)[17] which had not been observed despite a large number of previous, but powder XRD, studies[18–23].

In this paper, we present the synthesis of seven lanthanum hydrides, as well as two carbon-containing compounds, in the range of 50–176 GPa, all characterized by employing SCXRD. The unit cell parameters, space group, and positions of non-hydrogen atoms were unambiguously experimentally determined for all compounds, deduced to be $LaH_3$, $LaH_{-4}$, $LaH_{4+\delta}$, $La_4H_{23}$, $LaH_{6+\delta}$, $LaH_{9+\delta}$, $LaH_{10+\delta}$, LaC, and $LaCH_2$ through comparisons with structural analogues and their volume per lanthanum atom. Our results demonstrate the compositional and structural variety of hydrides in the La–H system and reveal regularities in the high-pressure rare-earth elements–hydrogen (RE–H) systems. Moreover, we establish that high-pressure syntheses employing high-temperature laser-heating can result in a significant sample heterogeneity that underpins difficulties in the interpretation of physical phenomena observed in the RE–H systems at low temperatures.

## Results

Three DACs with anvil culets of 80 μm were loaded with small lanthanum pieces along with paraffin oil ($C_nH_{2n+2}$) acting as a pressure-transmitting medium and a hydrogen reservoir. As demonstrated in recent works on metal hydrides[14,15,17,24–27], paraffin is an effective alternative to pure hydrogen for DAC synthesis experiments. In fact, paraffin is deemed a better choice than the more commonly used ammonia borane ($NH_3BH_3$) as it does not bring any additional elements into the experimental chamber, carbon being already present in the system due to the diamond anvils. The sample pressure was determined from the X-ray diffraction signal of the Re gaskets[28]—non-hydrogenated when using paraffin—and crosschecked with the diamond anvils' Raman edge[29]. Further details on the experimental and theoretical methods are described in the Supplementary Materials. Lanthanum and paraffin were compressed in three DACs at ambient temperature and laser-heated above 2000 K at pressures of 96, 106, 140, 150, 155, and 176 GPa (see Supplementary Table 1 for P–T

conditions and a list of the phases observed). The chemical reaction products were probed using synchrotron X-ray diffraction, and sample X-ray diffraction maps are provided in Supplementary Figs. 1–3.

A total of nine compounds were observed: seven lanthanum hydrides and two carbides. For all of them, the determined unit cell parameters, space groups, and atomic coordinates were determined based on the arrangement formed by the non-hydrogen atoms (i.e., lanthanum and carbon). These structures, for which Pearson symbols are provided, were solved and refined exclusively from the SCXRD data obtained from microcrystals. The lanthanum atoms in the seven lanthanum hydrides, $LaH_3$, $LaH_{-4}$, $LaH_{4+\delta}$, $La_4H_{23}$, $LaH_{6+\delta}$, $LaH_{9+\delta}$, and $LaH_{10+\delta}$, are arranged as shown in Fig. 1—with the hydrogen content in these phases, and therefore the assigned compound stoichiometry, being discussed afterward. The structural data are also summarized in Table 1 (full crystallographic data in Supplementary Tables 2–18).

The $LaH_3$ compound (Fig. 1a and Supplementary Table 2) was observed at 50 GPa. La atoms, forming a cubic close-packing (*ccp*), are located in the nodes of the *fcc* lattice (*cF4*, space group *Fm–3m*) with the unit cell parameter of 5.019(3) Å (V = 126.43(13) Å$^3$).

At 140 GPa, in the $LaH_{-4}$ solid (Fig. 1b and Supplementary Tables 3 and 4), the La atoms are arranged in an orthorhombic unit cell (*oC4*, *Cmcm* space group) with a = 2.949(4), b = 6.7789(19), and c = 4.7837(18) Å (V = 95.63(14) Å$^3$). This solid was also produced after further sample compression to 155 GPa and laser-heating.

Drawn in Fig. 1c, $LaH_{4+\delta}$ was observed after laser-heating at 140, 150, and 155 GPa (Supplementary Tables 5–7). La atoms are located in the nodes of the body-centered tetragonal unit cell (*tI2*, *I4/mmm* space group) with the lattice parameters of a = 2.9418(12) and c = 6.028(3) Å (V = 52.17(4) Å$^3$) at 140 GPa.

The $La_4H_{23}$ solid was obtained after laser-heating at four different pressures: 96, 106, 140, and 150 GPa (Fig. 1d and Supplementary Tables 8–11). The structure formed by the La atoms has a cubic symmetry (*cP8*, space group *Pm–3n*). At 150 GPa, it has a lattice parameter of 6.0722(8) Å (V = 223.89(5) Å$^3$).

The $LaH_{6+\delta}$ compound is the only one that was solely observed at a single pressure upon laser-heating: 150 GPa (Fig. 1e and Supplementary Table 12). The cubic structure formed by La atoms located in the nodes of the body-centered lattice (*cI2*, *Im–3m* space group) has a lattice parameter of a = 3.8710(10) (V = 58.01(3) Å$^3$).

Both the $LaH_{9+\delta}$ and the $LaH_{10+\delta}$ solids were observed at 140 and 176 GPa after laser-heating. In $LaH_{9+\delta}$ (Fig. 1f and Supplementary Tables 13 and 14), La atoms form a hexagonal close-packing (*hcp*) (*hP2*, space group $P6_3/mmc$) with the unit cell parameters a = 3.772(2) and c = 5.634(4) Å (V = 69.41(7) Å$^3$). In $LaH_{10+\delta}$ (Fig. 1g and Supplementary

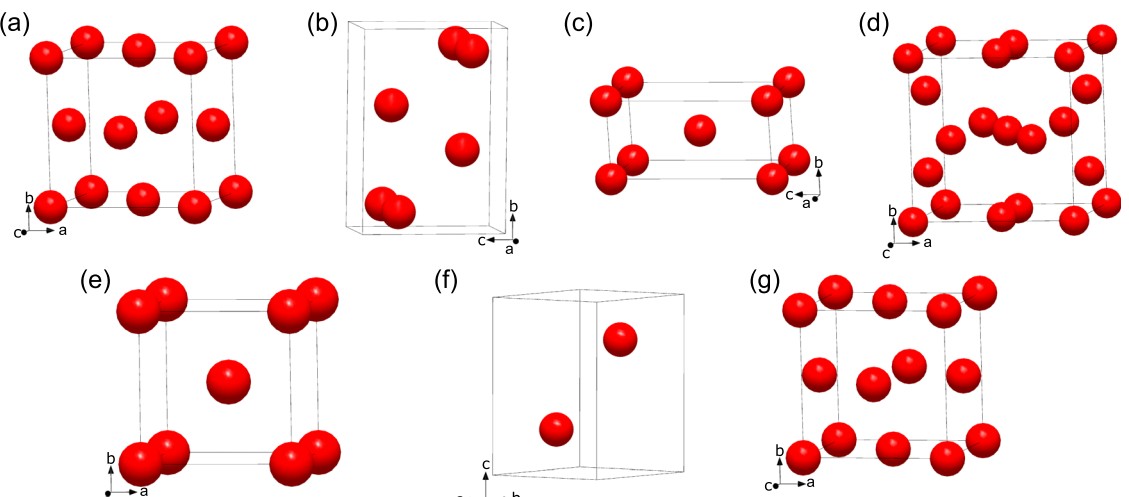

**Fig. 1 | Arrangement of lanthanum atoms in the lanthanum hydrides synthesized in this work. a** $LaH_3$; **b** $LaH_{-4}$; **c** $LaH_{4+\delta}$; **d** $La_4H_{23}$; **e** $LaH_{6+\delta}$; **f** $LaH_{9+\delta}$; **g** $LaH_{10+\delta}$. The red spheres represent lanthanum atoms.

**Table 1 | Selected crystallographic data for LaH₃, LaH₋₄, LaH₄₊δ, La₄H₂₃, LaH₆₊δ, LaH₉₊δ, and LaH₁₀₊δ**

| Compound | LaH₃ | LaH₋₄ | LaH₄₊δ | La₄H₂₃ | LaH₆₊δ | LaH₉₊δ | LaH₁₀₊δ |
|---|---|---|---|---|---|---|---|
| Pearson symbol | cF4 | oC4 | tI2 | cP8 | cI2 | hP2 | cF4 |
| Pressure (GPa) | 50 | 140 | 140 | 150 | 150 | 140 | 140 |
| Space group | Fm−3m | Cmcm | I4/mmm | Pm−3n | Im−3m | P6₃/mmc | Fm−3m |
| a (Å) | 5.019(3) | 2.949(4) | 2.9418(12) | 6.0722(8) | 3.8710(10) | 3.772(2) | 5.2233(14) |
| b (Å) | 5.019(3) | 6.7789(19) | 2.9418(12) | 6.0722(8) | 3.8710(10) | 3.772(2) | 5.2233(14) |
| c (Å) | 5.019(3) | 4.7837(18) | 6.028(3) | 6.0722(8) | 3.8710(10) | 5.634(4) | 5.2233(14) |
| V (Å³) | 126.43(13) | 95.63(14) | 52.17(4) | 223.89(5) | 58.01(3) | 69.41(7) | 142.51(7) |
| Z | 4 | 4 | 2 | 8 | 2 | 2 | 4 |
| V (Å³)/La atom | 31.61(3) | 23.91(1) | 26.09(2) | 27.986(6) | 29.01(2) | 34.71(4) | 35.63(2) |
| R$_{int}$ | 0.1391 | 0.0401 | 0.0166 | 0.0486 | 0.0674 | 0.023 | 0.0454 |
| R₁ (I ≥ 3σ) | 0.1224 | 0.0441 | 0.0400 | 0.0329 | 0.0490 | 0.0353 | 0.0495 |
| wR₂ (I ≥ 3σ) | 0.1257 | 0.0575 | 0.0478 | 0.0384 | 0.0620 | 0.0415 | 0.0612 |
| R₁ (all data) | 0.1256 | 0.0453 | 0.0400 | 0.0416 | 0.0490 | 0.0356 | 0.0498 |
| wR₂ (all data) | 0.1259 | 0.0576 | 0.0478 | 0.0390 | 0.0620 | 0.0415 | 0.0612 |

Pearson symbols refer to the structures formed by La atoms (hydrogen atoms are not accounted as their positions could not be determined from the experimental data). The full experimental and crystallographic data for each phase and the pressures at which they have been observed are provided in Supplementary Tables 2–16. The crystallographic data has been submitted to the CCDC database under the deposition numbers CSD 2196053–2196069.

Tables 15–16) the lanthanum atoms adopt a *ccp* arrangement (*cF*4, *Fm*−3*m* space group) with the unit cell parameters a = 5.2233(14) Å, V = 142.51(7) Å³.

Before addressing the hydrogen composition of these solids, it must be emphasized that great care was taken to verify that the seven observed La-H compounds are free of carbon; carbon atoms which could potentially originate from the diamond anvils or from the decomposed paraffin oil. Indeed, at the stage of the structure refinement the possibility to introduce carbon into the structure was always checked but led to the failure of the structure refinement. However, at 96 GPa, a previously unknown carbohydride, LaCH₂, was detected by SCXRD, and its structure was solved (see Supplementary Table 17 and Supplementary Fig. 4), whereas at 150 GPa, a previously unobserved lanthanum carbide LaC was obtained and its structure was also determined (Supplementary Table 18 and Supplementary Fig. 5). The diffraction signal of polycrystalline diamond could be detected throughout the sample chamber after sample laser-heating (see Supplementary Fig. 1), suggesting it to be a decomposition product of paraffin oil.

The LaH₃, LaH₄₊δ, La₄H₂₃, LaH₆₊δ, LaH₉₊δ, and LaH₁₀₊δ solids have, respectively, metal atom arrangements like in the known LaH₃[30] and in the predicted LaH₄[31], RE₄H₂₃ (RE = Eu[32,33], and in the Ba-H system[34]), REH₆ (RE = Y[4,35], Eu[33]), REH₉ (RE = Y[4], Ce[36], Pr[37,38], Nd[39], Eu[32,33]) and the established LaH₁₀[2,3,40]. It is interesting to note that the unit cells of LaH₋₄ and LaH₄₊δ are very similar to some suggested for La-D compounds[41], while La₄H₂₃ and LaH₆₊δ are likely to have been seen— but not identified as such—in previously reported powder X-ray diffraction data[40]. Density functional theory (DFT) calculations carried out for La hydrides assuming the hydrogen positions as in the above-mentioned prototypes predict LaH₄, LaH₆, LaH₉, and LaH₁₀ as dynamically (anharmonically) stable at their synthesis pressure (Supplementary Fig. 6). However, when calculating the equation of state (EoS) of these solids using DFT (Fig. 2a), the agreement with the experimental volume per lanthanum atom was found to be surprisingly poor —even when considering a ±10 GPa uncertainty in the experimental pressure. The exceptions to this were LaH₃ and La₄H₂₃ for which a reasonable match is obtained, aside from one point for La₄H₂₃ at 150 GPa. Indeed, the difference between the experimental and the calculated volume per lanthanum atom (Fig. 2b) is below 1 Å³ for the LaH₃ and La₄H₂₃ compounds (the 150 GPa point of La₄H₂₃ aside), but between +1.73 and +3.83 Å³ for the five other hydrides. Regarding the metal arrangement in LaH₋₄, to the best of our knowledge, it has

neither been experimentally observed nor predicted in hydrides. However, from its volume per lanthanum atom, its hydrogen content is expected to be in between that in LaH₃ and LaH₄₊δ, hence named LaH₋₄.

Strikingly, a volume per metal atom discrepancy between the experimental and calculated data for the hypothesized stoichiometry of a hydride is common in the literature. As shown in Supplementary Fig. 7, this is especially true for LaH₁₀−for which the largest number of independent studies were conducted−featuring a very wide range of volume per La atom with most points significantly differing from the calculated data. This is also the case for CeH₃[36], EuH₅ and EuH₆[33], PrH₄ and PrH₉[38], UH₃[42], and Ba₄H₂₃[34], for all of which a considerable fraction of the experimental volume per metal atom datapoints lie 1.5 Å³ or more from the corresponding calculated curve.

Three hypotheses to account for the aforementioned discrepancy between experimental and calculated volume differences can be considered: (a) The experimental uncertainty on both the measured pressure and volume. The experimental error on the volume is usually very small, typically much smaller than ±0.2 Å³[2-4,38,42]. Regarding the pressure uncertainty, it is often reported to be below ±5 GPa[2,3,42], even at pressures of 180 GPa. Moreover, laser-heating typically homogenizes stress within the sample chamber[43]. Such an uncertainty does not account for the difference with the calculated volumes. However, it is worth noting that some studies have reported a gap of up to 30 GPa between the pressure values measured by the H₂ vibron and the diamond Raman edge[3,4], the latter always being higher in pressure. Such a large error could indeed be responsible for the observed differences when the diamond Raman edge is employed as the sole pressure gauge. However, in all cases where pressure measurements are described and a large unit cell volume gap exists between DFT calculations and experiments[33,34,38,42], at least two pressure gauges were employed−often the diamond Raman edge along with an X-ray gauge. In this study, the difference between the pressure measured from the diamond anvils Raman edge and the Re diffraction signal was always found to be equal or below 10 GPa. The use of two pressure gauges greatly diminishes the likelihood of pressure measurements being responsible for the unit cell volume discrepancy.

(b) DFT calculations of RE hydride compounds are particularly demanding as thermal effects and the quantum nature of hydrogen significantly influence key properties, *e.g.* the obtained pressure, and can lead to different (usually higher symmetry) structures being thermodynamically most stable at a given *P*, *T* conditions[44,45].

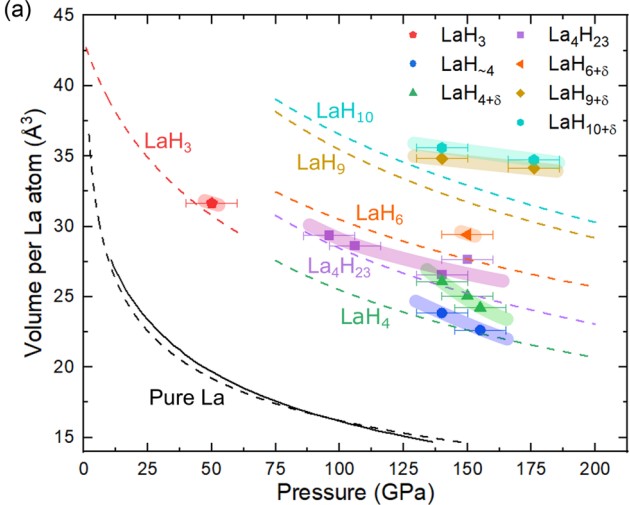

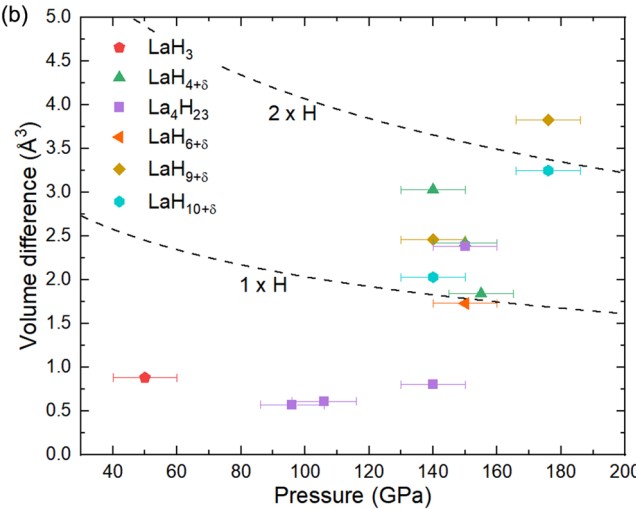

**Fig. 2 | Analysis and comparison of the experimental X-ray diffraction data with theoretical calculations. a** Unit cell volume per La atom as a function of pressure, plotted for the seven synthesized lanthanum hydrides. The solid symbols represent the experimental data obtained in this study, for which the error bars are the largest difference observed between the rhenium and the diamond pressure gauges (±10 GPa). The colored broad bands serve as guides to the eye and are drawn based on a 2nd-order Birch–Murnaghan equation of state (see Supplementary Table 19), when sufficient datapoints are available. These have no underlying physical meaning as they suffer from two main issues: (i) the EoS parameters were obtained from very few (often two) experimental datapoints and (ii) the hydrogen content, for a given lanthanum atom arrangement, is thought to vary, which renders inaccurate the EoS parameters. The dotted lines are the calculated equation of state for LaH₃ (red), LaH₄ (green), La₄H₂₃ (purple), LaH₆ (orange), LaH₉ (dark yellow) and LaH₁₀ (cyan), with structures based on the known RE-H compounds with an identical arrangement of metal atoms, i.e. REH₃[30], REH₄[31], RE₄H₂₃[32,33], REH₆[4,33,35], REH₉[4,32,33,36–39], and REH₁₀[2,3,40]. The full black line represents the experimental equation of state of pure lanthanum[48]. **b** The difference in volume per lanthanum atom between the experimental datapoints and the corresponding calculated EoS of (**a**). The two black dashed lines are the calculated volume of one and two hydrogen atoms, based on the EoS of atomic hydrogen[49]. Source data are provided as a Source Data file.

Furthermore, REs heavier than La require an advanced treatment of *f*-electrons (Ce, Eu, etc.) as well as taking into account spin-orbit coupling[32,39]. Also, the approximation employed for the exchange-correlation functional can lead to a pressure difference of the order of 10 GPa (e.g. LDA -10 GPa lower compared to PBE for examples with lanthanum hydrides, see Supplementary Table 20). Thermal or nuclear

quantum effects can also be a reason for the difference between the experimental and computational values. This is tested here using the temperature-dependent effective potential (TDEP[46,47], which includes anharmonic effects) to obtain a pressure correction for pressure-volume points of some lanthanum hydrides (Supplementary Fig. 8). An average volume per La atom difference of 2.2 Å³ was found for the LaH₄, LaH₆, LaH₉ and LaH₁₀ compounds between the experimental data and the computed TDEP results; still insufficient to explain the volume discrepancy or the difference in the slope of the LaH₉₊δ and LaH₁₀₊δ datapoints (Supplementary Fig. 8).

(c) The third possibility, of physical nature, is the variability of hydrogen content for a given or mildly distorted arrangement of the metal atoms arrangement, i.e. a given arrangement of metal atoms can have a range of hydrogen compositions. This variability would likely depend on pressure, temperature, and the number of hydrogen atoms available. There are a number of established examples of this phenomenon in the literature[3,36,38]. Among the most striking ones is the report of the formation of LaH₁₀ from LaH₃ embedded in H₂—both hydrides sharing the same *ccp* metal arrangement—simply by leaving the sample at -140 GPa for two weeks[3]. In another study, cerium was loaded in H₂ and compressed up to about 160 GPa at room temperature. Five phases were inferred to be produced based on powder X-ray diffraction measurements, sequentially forming CeH₃, CeH₃₊ₓ, CeH₄, CeH₉₋ₓ, and CeH₉, each time accompanied by a mild distortion of the Ce sublattice[36]. This hydrogen content variability provides a straightforward explanation to discrepancies between experimentally- and DFT-derived unit cell volumes: the hydrogen content is not the same, and therefore the arrangement of hydrogen atoms can differ substantially. This underlines the inadequacy of DFT calculations to determine a hydride's composition and full structure solely based on the arrangement of the metal atoms.

In the case of the here-synthesized lanthanum hydrides, the variability of H atoms justifies the assigned stoichiometry of LaH₃, LaH₄₊δ, La₄H₂₃, LaH₆₊δ, LaH₉₊δ, and LaH₁₀₊δ, some of these solids containing more hydrogen compared to the stoichiometry expected based on their La atoms' arrangement (Fig. 2). Estimates of the hydrogen content in the synthesized phases also can be proposed assuming an ideal mixing of the pure lanthanum[48] and atomic hydrogen[49] (Supplementary Fig. 4) as well as based on the DFT calculations with a hydrogen volume inferred from the calculated EoS of LaH₃, LaH₄, La₄H₂₃, LaH₆, LaH₉, and LaH₁₀ (Supplementary Fig. 7). Both approaches confirm a higher-than-expected hydrogen content, although the exact stoichiometry of these hydrides remains to be determined.

The difficulties of capturing, with DFT calculations, the flexibility of a given metal atoms' arrangement with regards to hydrogen content is especially alarming given the extreme reliance of experiments on these calculations. Indeed, these are very often used to assess the quantity and location of hydrogen atoms, crucial to determine the hydrides' full structural model as well as to predict and interpret high-temperature superconductivity. Further emphasizing this, high-temperature superconductivity was measured on samples assumed to be LaH₁₀ but found to have a unit cell volume 3.29 Å³ smaller than the volume expected based on the calculated EoS of LaH₁₀[40]. This roughly corresponds to two fewer hydrogen atoms, based on the EoS of atomic H at 180 GPa (1.67 Å³/H, see Fig. 2b)[49]. Obviously, whether with more or less hydrogen atoms, the structural model of this "LaH₁₀" compound, and therefore its critical temperature, should be significantly different.

With a total of seven solids found stable at high pressures, each with a different La:H ratio, La-H is a binary hydride system with one of the largest number of experimentally observed distinct compounds. The discovery of these solids also sheds light on regularities among synthesized hydrides of rare-earth elements Y, La, Ce, Pr, Nd, and Eu (Fig. 3) with respect to their metal atoms' arrangement. These can be

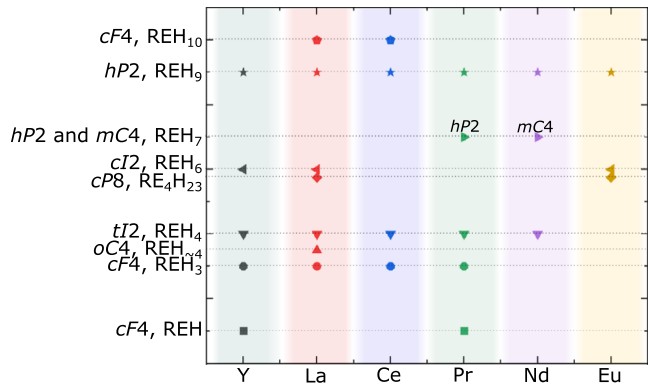

**Fig. 3 | Occurrence of a given metal atoms' arrangement in known hydrides of Y, La, Ce, Pr, Nd, and Eu**[4,32,33,37–39,50]. For clarity, the metal atoms' arrangements are labeled by both their Pearson symbol (excluding hydrogen atoms) as well as by the chemical composition typically assumed in the literature–but likely slightly different in view of this work. For REH$_7$, the known $hP2$ PrH$_7$[38] and $mC4$ NdH$_7$[39] have the same stoichiometry but distinct metal sublattices. Source data are provided as a Source Data file.

grouped as the following: LaH$_3$, YH$_3$ and PrH$_3$ ($cF4$); LaH$_{4+\delta}$, YH$_4$, CeH$_4$, PrH$_4$, and NdH$_4$ ($tI2$); La$_4$H$_{23}$ and Eu$_4$H$_{23}$ ($cP8$); LaH$_{6+\delta}$ and YH$_6$ ($cI2$); LaH$_{9+\delta}$, YH$_9$, CeH$_9$, PrH$_9$, NdH$_9$, and EuH$_9$ ($hP2$); LaH$_{10+\delta}$ and CeH$_{10}$[4,32,37–39,50] ($cF4'$). Considering the chemical similarities of these rare-earth elements, their propensity to forming isostructural compounds is expected. Perhaps what is more noteworthy is that not all metal atoms' arrangement appear to be common to all RE-H systems. For example, the $tI2$ arrangement was observed in all systems with the exception of EuH. It is likely that it could be formed, but was missed in previous powder XRD studies[32,33]. In that regard, SCXRD measurements have proven, both here and previously[17], to be particularly powerful compared to powder XRD. Without a structural refinement based on SCXRD data, the presence of non-hydrogen light elements, like carbon or boron and nitrogen (in experiments with ammonia borane in the sample chamber), in the structure of hydrides could hardly be ruled out. As it stands, the vast majority of rare-earth hydrides were "identified" without any sort of structural refinements, instead heavily relying on theoretical calculations, which, as here shown in the case of lanthanum hydrides, are not always capable of predicting all compounds that may be synthesized.

The results presented in this study have profound implications for the interpretation of the results of superconductivity measurements in RE-H systems. In the range of 140 to 155 GPa, seven La-H phases, as well as LaC, have been demonstrated to be synthesizable (see Supplementary Table 1), including all but LaH$_{6+\delta}$ to be simultaneously present in DAC #1 at 140 GPa. The presence of multiple phases in a given sample at a given pressure hinders the quantitative interpretation of resistivity and magnetic susceptibility measurements aimed at determining the superconducting temperature. In particular, these five previously unobserved La-H compounds are likely to be the cause of the drops in resistance at temperatures other than 250 K—as well as the many unidentified diffraction peaks—found in the previous studies[3]. Likewise, the results of X-ray spectroscopy investigations, recently suggested for the characterization of the hydrogen atoms in hydrides[6], would also be affected by a strong sample inhomogeneity. The presence of these multiple phases could even lead to the false determination of the $T_c$ value, as it increases the probability of the formation of metallic conduction paths, as described by Hirsch and Marsiglio[7]. In this context, it can be pointed out that if experimental measurements and theoretical calculations for the determination of the $T_c$ in superhydrides[2,3,45] are in fact accurate, the LaH$_{6+\delta}$ and LaH$_{9+\delta}$ solids could prove to be promising targets as high $T_c$ materials. Indeed, YH$_6$

and YH$_9$, with the same metal arrangement as in LaH$_{6+\delta}$ and LaH$_{9+\delta}$, were suggested to have $T_c$ values of 220 K and 243 K, respectively, at pressures of 183 and 201 GPa[4].

The results presented here suggest a wide variability of the hydrogen content for a given structure formed by La atoms. The "LaH$_{10}$" compound is a prime example, with a hydrogen composition that would be expected to vary by ±2 based on the comparison between the experimental and calculated datapoints. Moreover, we unveiled crystal-chemical regularities common for various RE-H systems: all La hydrides studied in this work, with the exception of LaH$_{-4}$, were found to adopt the same La arrangements like those previously known in other RE-H systems. The detection of seven La-H compounds in the pressure range of 140–176 GPa–precisely in which superconducting samples were reported–points out the significant difficulties in having single-phase samples that are necessary for a reliable assessment of physical properties of materials, including superconductivity. Our study should promote the use of SCXRD on microcrystalline samples as an essential tool to characterize hydrides given its demonstrated ability to detect phases otherwise missed by powder XRD analysis. In particular, this approach should be employed to characterize lanthanum-compressed and laser-heated DACs along with ammonia borane to verify that no nitrides or borides other than BN are produced. Further investigations of the LaH$_{6+\delta}$ and LaH$_{9+\delta}$ compounds are of interest, as they potentially could be superconducting at high temperatures.

## Methods

### Experimental

Three BX90-type screw diamond anvil cells (DACs) were equipped with diamonds of culet sizes of 80 μm and rhenium was employed as the gasket material. The samples, composed of lanthanum embedded in paraffin oil, were prepared in one of two ways. One way was with a lanthanum ingot (99.9% purity), purchased from Sigma Aldrich, that was kept in an argon glovebox. When needed, micrometer-sized pieces of La were scratched off the lanthanum ingot in the glovebox and put in paraffin oil (C$_n$H$_{2n+2}$) immediately after being taken out of the glovebox, preventing a reaction with air, and then loaded in the DACs. For the second approach, lanthanum pieces already under paraffin oil (C$_x$H$_{2+x}$) was purchased. The lanthanum was cut down to the appropriate micrometer-size pieces in paraffin oil and directly transferred in the DACs. Paraffin was used as a pressure-transmitting media as well as a hydrogen reservoir, as it was successfully done for the synthesis of several other hydrides[14,15,17,24–27]. The sample pressure was determined from the X-ray diffraction signal of the Re gaskets[28]—never hydrogenated using paraffin—and crosschecked with the diamond anvils' Raman edge[29].

Lanthanum pieces of various sizes were loaded in the DACs so as to cover a wide lanthanum-to-hydrogen ratio. The samples were compressed at ambient temperature and laser-heated above 2000 K at pressures of 96, 106, 140, 150, 155, and 176 GPa. The double-sided YAG laser-heating of the samples was performed mainly at the GSECARS beamline of the APS and partially at the Bayerisches Geoinstitut[51], with lanthanum acting as the laser absorber. Both system are quite similar and both allowed a focused (circular) laser spot of about 8 μm in diameter and enter the diamond anvil at perpendicular incidence. A flat-top beam is used, which provides a roughly equal number of photons per area within that 8 μm diameter. In all cases, temperatures were measured using the samples' thermal radiation[52], although very high-temperature spikes of short during could usually not be measured. The samples were laser-heated until sample recrystallization, observed through the appearance of new sharp reflections, to a maximum measured temperature below 3200(200) K (see Supplementary Table 1).

The samples were mainly characterized by single-crystal (SCXRD) and powder X-ray diffraction (XRDp) measurements. The SCXRD was

performed on very small, just few-micron size of single crystals, for which a special approach to the high-pressure XRD data acquisition and analysis was recently developed[53]. This approach was demonstrated on many systems[54–56]. The X-ray diffraction data were acquired at the P02.2 and GSECARS beamlines of PETRA III and the APS, respectively. At the P02.2 beamline, a Perkin Elmer XRD 1621 detector was employed with an X-ray beam of wavelength of $\lambda = 0.2901$ Å, focused down to about $2 \times 2\,\mu m^2$. At the GSECARS beamline, a Pilatus CdTe 1 M detector was used along with an X-ray beam focused down to $3 \times 3\,\mu m^2$, with a wavelength of $\lambda = 0.2952$ Å. On the polycrystalline samples, a full X-ray diffraction mapping of the experimental cavity was performed after each laser-heating in order to identify the most promising sample positions for a single-crystal data collection. On the locations where the most intense single-crystal reflections were detected, single-crystal data were acquired in step-scans of 0.5° from −38° to +38°. The CrysAlis$^{Pro}$ software[57] was utilized for the single-crystal data analysis. To calibrate an instrumental model in the CrysAlisPro software, i.e., the sample-to-detector distance, detector's origin, offsets of goniometer angles, and rotation of both X-ray beam and the detector around the instrument axis, we used a single crystal of orthoenstatite ($(Mg_{1.93}Fe_{0.06})(Si_{1.93}, Al_{0.06})O_6$), *Pbca* space group, $a = 8.8117(2)$, $b = 5.18320(10)$, and $c = 18.2391(3)$ Å. The same calibration crystal was used at all the beamlines. The analysis procedure in the CrysAlis$^{Pro}$ software includes the peak search, the removal of the diamond anvils' parasitic reflections and saturated pixels of the detector, finding reflections belonging to a unique single crystal, the unit cell determination, and the data integration. The crystal structures were then solved with SHELXT structure solution program[58] using intrinsic phasing and refined within the JANA2006 software[59]. CSD 2196053–2196069 contain the supplementary crystallographic data for this paper. These data can be obtained free of charge from FIZ Karlsruhe via www.ccdc.cam.ac.uk/structures. Powder X-ray diffraction was also performed to verify the chemical homogeneity of the samples and the data integrated with Dioptas[60].

## Computational details

Kohn-Sham density functional theory (DFT) based structural relaxations and electronic structure calculations were performed with the QUANTUM ESPRESSO package[61–63] using the projector augmented wave approach[64]. We used the generalized gradient approximation by Perdew–Burke–Ernzerhof (PBE) for exchange and correlation[65], for which the 4d and lower electrons of lanthanum are treated as scalar-relativistic core states. Convergence tests with a threshold of 1 meV per atom in energy and 1 meV/Å per atom for forces led to a Monkhorst-Pack[66] k-point grid of $24 \times 24 \times 24$, $24 \times 24 \times 12$, $8 \times 8 \times 8$, $24 \times 24 \times 24$, and $24 \times 24 \times 12$ for $LaH_3$, $LaH_4$, $La_4H_{23}$, $LaH_6$, $LaH_9$, and $LaH_{10}$, respectively. For all phases, a cutoff for the wavefunction expansion of 80 Ry for wavefunction and 640 Ry for the density with 0.01 Ry Marzari–Vanderbilt smearing[67].

We performed variable-cell relaxations (lattice parameters and atomic positions) on all experimental structures to optimize the atomic coordinates and the cell vectors until the total forces were smaller than $10^{-4}$ eV/Å per atom and the deviation from the target pressure was below 0.1 GPa.

Equation of state (EoS) calculations are performed via variable-cell structural relaxations between 75 and 200 GPa for all lanthanum hydrides with the exception of LaH$_{-4}$ (see Fig. 2a), for which the complete crystal structure is unknown and $LaH_3$ (see below). We fit a third-order Birch–Murnaghan EoS to the energy-volume points, calculate $P(V)$, and benchmark versus the target pressure of the relaxations to ensure convergence. The crystal structure of all computationally investigated lanthanum hydrides is preserved without constraining their structures to the experimentally determined space group, even if all phases but $LaH_4$ feature negative modes in the harmonic approximation. Calculations on $LaH_3$ were performed

between 0 and 200 GPa, and it was found that it loses symmetry transitioning from space group *Fm−3m* (#225) to *R−3m* (#166) starting from P ≳ 70 GPa. The resulting EoS for the high-symmetry *Fm−3m* $LaH_3$ is in relatively good agreement with our experimental point at 50 GPa, while the *R−3m* $LaH_3$ EoS is in good agreement with points by Drozdov et al.[3].

DFT computations used as a basis for the finite-temperature calculations (see below and Supplementary Fig. 8) have been performed using the FHI-aims code[68,69] in supercells with orthogonal lattice vectors of at least 5 Å length using the following DFT parameters described below.

As explained in the main text, the choice of the exchange-correlation (xc) functional when using DFT significantly affects the static calculated pressures observed in lanthanum hydrides. To estimate the effect, we computed the static pressure in a $LaH_4$ cell using four commonly used xc-functionals, the local-density approximation (LDA) and three functionals of the generalized gradient approximation (GGA) family, with the FHI-aims code[68,69]. The results are listed in Supplementary Table 2. While LDA[70] agrees quite well with PBEsol[71] and am05[72], the difference to PBE[65] is about 7 to 10 GPa. While we use PBE for the rest of the calculations to match with the existing literature[45], we note that it seems to yield larger pressures than the other commonly employed functionals. We also investigated the influence of the basis sets used in FHI-aims[68]. We checked light and tight default basis sets, which yield a difference of 0.7 GPa in the structure studied above. We, therefore, conclude that the error when using light default basis sets is negligible compared to, e.g., the choice of the xc-functional, and use light basis sets for all calculations in the following.

**Finite-temperature simulations with TDEP.** Finite-temperature properties have been modeled in the framework of temperature-dependent effective potentials (TDEP)[46,47] using self-consistent sampling[73], with expressions for the pressure similar to those described in refs. [74], [75]. In TDEP, the nuclear system is described by a harmonic Hamiltonian in the canonical ensemble (NVT):

$$H^{TDEP}(\{\boldsymbol{u},\boldsymbol{p}\};V,T) = U_0 + \sum_i \frac{p_i^2}{2m_i} + \frac{1}{2}\sum_{ij,\alpha\beta} \Phi(V,T)_{ij}^{\alpha\beta} u_i^\alpha u_j^\beta \quad (1)$$

parametrized at a volume V and temperature T as described in refs. [46], [47]. Here, $\boldsymbol{u} = (u_1,..., u_N)$ are displacements from the reference positions for $N$ atoms, and likewise $\boldsymbol{p} = (p_1,..., p_N)$ are their momenta. i, j are atom indices while α, β label Cartesian coordinates. $m_i$ is the mass of atom i and $U_0$ is the baseline energy which enforces $\langle H^{TDEP}\rangle_T = \langle H^{DFT}\rangle_T$ with the canonical average $\langle\cdot\rangle_T$ (we omit N and $V$ in the thermodynamic average for clarity), and $\Phi(V, T)$ are the effective harmonic force constants at the $V$, $T$ conditions of interest.

The harmonic force constants $\Phi$ can be used to create displacements $\{\boldsymbol{u}\}$ that correspond to the harmonic canonical ensemble defined by the harmonic Hamiltonian in Eq. (1) corresponding to the scheme outlined in ref. 76. These samples can be used to create new input data (forces) for parametrizing the TDEP Hamiltonian as introduced in ref. 73. This procedure can be repeated self-consistently from an initial guess as explained in detail in the appendix of ref. 77, resulting in stochastic temperature-dependent effective potentials (sTDEP). The potential pressure at finite temperature, $P^{pot}(T)$, is estimated by computing the DFT pressure, $P^{DFT}$ in samples created from the effective harmonic model[73,76],

$$P^{pot}(T) = \langle P^{DFT}\rangle_T \quad (2)$$

where $\langle\cdot\rangle_T$ denotes an average over the samples. To estimate the kinetic contribution to the pressure, we use that in the harmonic approximation, the kinetic pressure equals the kinetic energy, i.e., half the

effective harmonic energy in the samples, modulo a volume-dependent prefactor in supercells of volume $V$[74,75]:

$$P^{kin} = \frac{1}{3V} \left\langle H^{TDEP} \right\rangle_T \qquad (3)$$

The total pressure, including temperature and nuclear quantum effects (through the sampling $\langle \cdot \rangle_T$), is then given by

$$P(T) = P^{kin}(T) + P^{pot}(T) \qquad (4)$$

Sampling was performed self-consistently with steps using an increasing number of samples. The convergence with respect to the number of self-consistent steps is material dependent, and can be very fast with only 16 supercells in the final step for the more harmonic $LaH_4$, or more involved. In the current case, it was never necessary to go beyond 64 samples in the final step, even in the much more strongly anharmonic $LaH_9$.

The chosen supercells were rectangular with at least 5 Å side length, resulting in supercell sizes up to 180 atoms. This is in line with earlier studies showing a rapid convergence of force constants with supercell size in LaH due to the short-range nature of the interatomic forces in metallic systems[45].

## Data availability

The details of the crystal structure investigations may be obtained from FIZ Karlsruhe, 76344 Eggenstein-Leopoldshafen, Germany (fax: +49-7247-808-666; e-mail: crysdata@fiz-karlsruhe.de) on quoting the deposition numbers CSD 2196053–2196069. All other datasets generated during and/or analyzed during the current study are available from the corresponding author upon reasonable request. Source data are provided with this paper.

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

## Acknowledgements

The authors acknowledge the Deutsches Elektronen-Synchrotron (DESY) and the Advance Photon Source (APS) for provision of beamtime at the P02.2 and GSECARS beamlines, respectively. D.L. thanks the Alexander von Humboldt Foundation, the Deutsche Forschungsgemeinschaft (DFG, project LA-4916/1-1), and the UKRI Future Leaders Fellowship (MR/V025724/1) for financial support. N.D. and L.D. thank the Federal Ministry of Education and Research, Germany (BMBF, grant no. 05K19WC1) and the DFG (projects DU 954–11/1, DU 393–9/2, and DU 393–13/1) for financial support. B.W. gratefully acknowledges funding by the DFG in the framework of the research unit DFG FOR2125 and within projects WI1232 and thanks BIOVIA for support through the Science Ambassador program. N.D. and I.A.A. also thank the Swedish Government Strategic Research Area in Materials Science on Functional Materials at Linköping University (Faculty Grant SFO-Mat-LiU No. 2009 00971). Support from the Knut and Alice Wallenberg Foundation (Wallenberg Scholar grant no. KAW-2018.0194), the Swedish e-science

Research Center (SeRC), and the Swedish Research Council (VR) grant no. 2019-05600 is gratefully acknowledged. F.K. also acknowledges support from the Swedish Research Council (VR) program 2020-04630, and the Swedish e-Science Research Centre (SeRC). Computations were performed on resources provided by the Swedish National Infrastructure for Computing (SNIC) at the PDC Centre for High-Performance Computing (PDC- HPC) and the National Supercomputer Center (NSC). For the purpose of open access, the authors have applied a Creative Commons Attribution (CC BY) licence to any Author Accepted Manuscript version arising from this submission.

## Author contributions

D.L., L.D., and N.D. designed the work. D.L. and L.D. prepared the high-pressure experiments. D.L., A.A., S.K., T.F., S.C., K.G., and V.P. performed the synchrotron X-ray diffraction experiments. D.L. and L.D. processed the synchrotron X-ray diffraction data. Fl.T., B.W. F.K., V.M., and I.A.A. performed the theoretical calculations. D.L., Fl.T., T.M., and L.D. contextualized the data interpretation. D.L., Fl.T., L.D., N.D., and I.A.A. prepared the first draft of the manuscript with contributions from all other authors. All the authors commented on successive drafts and have given approval to the final version of the manuscript.

## Competing interests

The authors declare no competing interests.
