## [Peer Review File · Nature Communications]

High-Pressure Synthesis of Seven Lanthanum Hydrides with a Significant Variability of Hydrogen ContentREVIEWER COMMENTS

Reviewer #1 (Remarks to the Author):

This manuscript reports a fascinating and timely investigation of structural diversity in the lanthanum hydrogen system at megabar pressures. The importance in this ostensibly rather technical study lies in the fact that multiple groups have reported close to room temperature superconductivity in what has been supposed to be LaH₁₀ at similar pressures. The key strength of this work lies in the experimental analysis of what turns out to be a complex and heterogenous system. It is supported by some theoretical computations, but it is likely the future dedicated studies will be more enlightening. The authors recognise this, and the focus of the work is appropriately on the experimental results. The prime role of the computational work is to highlight the limitations and challenges inherent in them. True state of the art computational predictions, at high levels of theory in particular treating the quantum nature of the proton are extremely challenging and out of the scope of the current work.

The most important message of the manuscript is a cautionary one - it is clear that despite theory having lead the experimenters to these systems, and signatures of superconductivity having been measured, there is much remaining to be understood. At the same time, the authors demonstrate that modern high pressure synthesis methods provide access to the a wide range of compositions and structures. This gives some hope for future progress in this field, given the vast number of structures theoretically predicted, but so far not synthesised.

The La₄H₂₃ structure is fascinating, and in addition to having been found in Eu, it has been studied in some detail in Ba at lower pressures. See:

Binns, Jack, Miguel Martinez-Canales, Bartomeu Monserrat, Graeme J. Ackland, Philip Dalladay-Simpson, Ross T. Howie, Chris J. Pickard, and Eugene Gregoryanz. "Synthesis of Weaire-Phelan Barium Polyhydride." *J. Phys. Chem. Lett.* 12, no. 20) (2021).

It is looking like it may be ubiquitous in the hydrides.

As mentioned, the main focus of this work is not the theory, but it would have been nice to compute a revised convex hull of the system highlighting the newly synthesised phases in the context of those measured or predicted previously.

The provision on the crystal structure in cif file format is greatly appreciated.

Overall, I recommend that this manuscript is published. It will give the community food for thought, and certainly provoke follow on studies.

Reviewer #2 (Remarks to the Author):

This is an interesting paper, that could potentially shed important light onto the question whether or not the hydrides under pressure are standard superconductors. I have several questions though, I hope the authors will answer at least some.

(1) It seems to me that all the latest studies of superconductivity in hydrides under pressure use ammonia borane, while these authors use paraffin instead, arguing it is better because it does not bring in additional elements (B and N). However, if the purpose is to shed light on the experiments detecting superconductivity, why wasn't ammonia borane used? How do we know that the hydrides synthesized will be the same or similar with ammonia borane and paraffin? And if paraffin is better, why don't the researchers looking for superconductors use it?

(2) I couldn't find detailed information on how the laser heating was performed. I understand that for the superconductivity experiments they traverse a 5- μ m-diameter laser spot horizontally and vertically across the diamond culets. Is the procedure the same here? How large is

the laser spot?

(3) The authors say that they find up to seven different compounds in the DAC. It would be useful to have more detailed information: can they estimate the proportion of the different compounds? their spatial distribution? is a given compound only in one region or spread out in little pieces throughout the DAC?

(4) The resistive transitions shown in Refs. [2] and [3] are rather narrow, 4K-10K in Ref. [2], ~20K in Ref. [3]. It seems to me the authors are suggesting that because the samples in the DAC are expected to have compounds with much lower H content than LaH₁₀, and correspondingly lower T_c's or no T_c's, this should give rise to much broader resistive transitions than observed. It would be useful if the paper would expand on this point, if possible quantitatively, if not at least qualitatively.

Reviewer #3 (Remarks to the Author):

The study on compressed La-H system has attracted great attention since the finding of near-room superconductivity in LaH₁₀ at a pressure of ~180 GPa that was actually stimulated by the theoretical prediction. The following experiments also found other phases or stoichiometries of La-H system. The further research on these remarkable systems remains required to clarify the detailed compositions and structures. In this manuscript, these authors carried out the single-crystal X-ray diffraction investigation on lanthanum hydrides at a pressure range of 50-180 GPa, in combination with density functional theory calculations. They found several stoichiometries of lanthanum hydrides under high pressure. Among these newly found phases, as I thought, the LaH₆ and LaH₉ are quite interesting, since both are predicted to be unstable in the previous studies. Moreover, the authors also found other interesting compounds like LaCH₂, which will help to if carbon participated in the reaction between the metals and hydrogen in the future experiment. These results are new and look reasonable. Therefore, I could recommend the publication of this manuscript after considering below comments and suggestions.

1. In Figure S5, what temperature is used for these TDEP simulations? How many time step and supercell size in the TDEP simulations.
2. For LaH₆, LaH₉ and LaH₁₀ systems, the authors could provide the uncertainty for hydrogen stoichiometry, while not for La₄H₂₃. Could authors provide a reason or make pertinent discussion in the manuscript on that.
3. Could the authors do the density of states simulations for YH₆, YH₉, LaH₆ and LaH₉, which will help make discussions by comparing the estimated T_c of LaH₆ and LaH₉ with YH₆ and YH₉.
4. In Fig. 2a, the authors may should discuss the equation of states for LaH₉ and LaH₁₀ that is quite flat comparing with LaH₄ and La₄H₂₃.
5. The authors should carefully check all the references. For example, the page number is incorrect for Ref. 9, 43, and 70.

NCOMMS-22-32433

Response to referees' comments:

To better structure our response to the referees' reports, each of their comments has been copied and our response to each of these comments is written in blue. When changes were made to the manuscript, the original statement from our paper is copied and followed by the updated text. The same changes are also highlighted in the marked up manuscript file.

Report of Reviewer #1

This manuscript reports a fascinating and timely investigation of structural diversity in the lanthanum hydrogen system at megabar pressures. The importance in this ostensibly rather technical study lies in the fact that multiple groups have reported close to room temperature superconductivity in what has been supposed to be LaH10 at similar pressures. The key strength of this work lies in the experimental analysis of what turns out to be a complex and heterogenous system. It is supported by some theoretical computations, but it is likely the future dedicated studies will be more enlightening. The authors recognise this, and the focus of the work is appropriately on the experimental results. The prime role of the computational work is to highlight the limitations and challenges inherent in them. True state of the art computational predictions, at high levels of theory in particular treating the quantum nature of the proton are extremely challenging and out of the scope of the current work.

The most important message of the manuscript is a cautionary one - it is clear that despite theory having lead the experimenters to these systems, and signatures of superconductivity having been measured, there is much remaining to be understood. At the same time, the authors demonstrate that modern high pressure synthesis methods provide access to the a wide range of compositions and structures. This gives some hope for future progress in this field, given the vast number of structures theoretically predicted, but so far not synthesised.

The Reviewer's short summary truly captures the essence of our manuscript. We thank the Reviewer for his/her very positive assessment of our work.

The La4H23 structure is fascinating, and in addition to having been found in Eu, it has been studied in some detail in Ba at lower pressures. See:

Binns, Jack, Miguel Martinez-Canales, Bartomeu Monserrat, Graeme J. Ackland, Philip Dalladay-Simpson, Ross T. Howie, Chris J. Pickard, and Eugene Gregoryanz. "Synthesis of Weaire-Phelan Barium Polyhydride." J. Phys. Chem. Lett. 12, no. 20) (2021).

It is looking like it may be ubiquitous in the hydrides.

Indeed, this structure-type does seem to be quite common. We have added this reference to our manuscript.

As mentioned, the main focus of this work is not the theory, but it would have been nice to compute a revised convex hull of the system highlighting the newly synthesised phases in the context of those measured or predicted previously.

As requested by the Reviewer, we have calculated a revised convex hull of the La-H system at 140 GPa, which includes the newly synthesized phases as well as those previously calculated. For these calculations, we have assumed that the hydrogen atoms' quantity and position in our novel La-H compounds are the same as in the known rare-earth hydrides (RE-H) structure types; i.e. REH₃, REH₄, RE₄H₂₃, REH₆, REH₉ and REH₁₀ (see caption of Figure 3 for references to these structure types).

As shown below, the LaH₄, LaH₆, LaH₉ and LaH₁₀ phases lie on the convex hull, together with the phases discussed in reference [29] (now ref [31]; i.e. LaH₂, LaH₃ (space group 63, *Cmcm*) and LaH₈). La₄H₂₃ is only slightly above the hull. At 140 GPa, *R-3m* LaH₃ (space group 166) is significantly higher in enthalpy than the *Cmcm* space group version presented in reference [29] (now [31]). In our experiments, we only observed *Fm-3m* LaH₃ at 50 GPa and not at higher pressures.

However, we want to emphasize that, as detailed in the manuscript, we have not experimentally determined the position and quantity of the hydrogen atoms in our new La-H compounds. Because of this, and the fact that there is a discrepancy between the experimental and the theoretical unit cell volumes when assuming common RE-H structure types, we prefer not to include this convex hull in our manuscript.

The provision on the crystal structure in cif file format is greatly appreciated.

Overall, I recommend that this manuscript is published. It will give the community food for thought, and certainly provoke follow on studies.

We thank the Reviewer for recommending our manuscript for publication.

Report of Reviewer #2

This is an interesting paper, that could potentially shed important light onto the question whether or not the hydrides under pressure are standard superconductors.

We thank the Reviewer.

I have several questions though, I hope the authors will answer at least some.

(1) It seems to me that all the latest studies of superconductivity in hydrides under pressure use ammonia borane, while these authors use paraffin instead, arguing it is better because it does not bring in additional elements (B and N). However, if the purpose is to shed light on the experiments detecting superconductivity, why wasn't ammonia borane used?

The purpose of this manuscript was to explore the lanthanum-hydrogen compositional landscape around 150 GPa. In reference [3], mainly H₂ was used as the H source, with each sample having very different ratios between H and La atoms in the sample chamber, resulting in the formation of both H-poor and H-rich La-H compounds (see Extended Data Figure #2 and #5 of [3]). Carbon (from the anvils) is anyhow part of the chemical system investigated, thus the choice of paraffin is natural. In an effort to produce similar phases as observed in reference [3]—where both resistivity and XRD data suggest the formation of La-H compounds with various H content—it was important for us to use a precursor that allows for both H-poor and H-rich compounds (as well as potentially carbon bearing) to be formed. We feel that paraffin very well serves that purpose, as demonstrated by synthesizing both H-poor (*e.g.* LaH₃) and H-rich (*e.g.* LaH_{10+δ}) phases, and LaCH₂.

Moreover, we have clear experimental evidences that ammonia borane reacts with lanthanides forming rather complex compounds, and these results will soon be presented elsewhere. This is another reason why in the present work we avoid NH₃BH₃ as a hydrogen source.

How do we know that the hydrides synthesized will be the same or similar with ammonia borane and paraffin?

As written above, our objective was to explore the La-H compositional landscape in order to reproduce and characterize the many La-H compounds evidenced in reference [3] (*i.e.* Extended Data Figure #2 and #5), which have both H-poor and H-rich La-H phases. Our choice of using paraffin to accomplish this purpose is justified as we have produced LaH₃ and LaH₁₀—previously observed in samples with H₂ and ammonia borane as H sources—along with 5 phases with intermediate H-content.

And if paraffin is better, why don't the researchers looking for superconductors use it?

We are not comfortable speculating on why other research groups do not use paraffin. We can only restate that we believe paraffin to be a better choice since it does not bring in additional elements in the experimental chamber and allows to produce both H-rich and H-poor compounds, as desired in our experiments.

(2) I couldn't find detailed information on how the laser heating was performed. I understand that for the superconductivity experiments they traverse a 5- μm -diameter laser spot horizontally and vertically across the diamond culets. Is the procedure the same here? How large is the laser spot? The double-sided laser-heating system at the GSECARS beamline of the APS—setup mainly used in this study—allows focusing at the same sample position from the both sides of the pressure chamber. The focused (circular) laser spot is of about 8 μm in diameter and enters the diamond anvils at perpendicular incidence. A flattop beam is used, which provides a roughly equal number of photons per area within that 8 μm diameter. A very similar setup was employed at the Bayerisches Geoinstitut. Detail previously absent were added to the experimental method's description of the manuscript.

(3) The authors say that they find up to seven different compounds in the DAC. It would be useful to have more detailed information: can they estimate the proportion of the different compounds? their spatial distribution? is a given compound only in one region or spread out in little pieces throughout the DAC?

In response to the Reviewer's comment, two additional figures were added to the supplementary materials (see below). These figures are sample X-ray diffraction maps at 140 and 150 GPa after laser-heating, adding to the map already provided in Figure S3 (now Figure S1) for a sample at 176 GPa. At 140 GPa, the LaH_{-4} , $\text{LaH}_{4+\delta}$, La_4H_{23} , $\text{LaH}_{9+\delta}$, and $\text{LaH}_{10+\delta}$ phases were observed while at 150 GPa the $\text{LaH}_{4+\delta}$, La_4H_{23} , $\text{LaH}_{6+\delta}$, and LaC phases were detected. The X-ray diffraction maps were constructed using a characteristic X-ray diffraction peak of each phase which had minimal overlap with peaks of other compounds. The intensity of the chosen peak for each phase was used to plot the figures, where little intensity means faint colors and bright colors mean high intensity. Such maps are likely the only way to provide an insight into the proportion and spatial distribution of the La-H phases. It must be emphasized that these maps only provide qualitative information, on account of preferential orientation of the crystallites, the X-ray beamsizes ($2 \times 2 \mu\text{m}^2$), uneven laser-heating, and pressure, temperature and chemical gradients.

Figure S2: Sample X-ray diffraction map at 140 GPa, collected as 11x11 images with 2 μm steps. For each phase, an X-ray diffraction line not significantly overlapping with that of another phase was selected. For $\text{LaH}_{10+\delta}$, $\text{LaH}_{9+\delta}$, La_4H_{23} , $\text{LaH}_{4+\delta}$ and $\text{LaH}_{\sim 4}$, the 6.48, 6.00, 5.69, 6.26 and 8.14° 2θ peaks ($\lambda = 0.29521 \text{ \AA}$) were selected, respectively. Each panel shows the XRD map with one phase visible, with the color's intensity being proportional to the intensity of its X-ray diffraction line. To allow comparison between images, all intensities were normalized to the most intense peak out of any phases (*i.e.* $\text{LaH}_{9+\delta}$). For $\text{LaH}_{4+\delta}$, the intensity of the chosen diffraction line is so weak that it is barely visible. The sample region featuring some $\text{LaH}_{4+\delta}$ is encircled in yellow. Due to the compounds' preferred orientation, these maps can solely be qualitatively interpreted.

Figure S3: Sample X-ray diffraction map at 150 GPa, collected as 9x9 images with 2 μm steps. For each phase, an X-ray diffraction line not significantly overlapping with that of another phase was selected. For $\text{LaH}_{6+\delta}$, La_4H_{23} , $\text{LaH}_{4+\delta}$ and LaC , the 16.40, 6.82, 6.51, and 13.77° 2θ peaks ($\lambda = 0.29521 \text{ \AA}$) were selected, respectively. Each panel shows the XRD map with one phase visible, with the color's intensity being proportional to the intensity of its X-ray diffraction line. To allow comparison between images, all intensities were normalized to the most intense peak out of any phases (*i.e.* $\text{LaH}_{4+\delta}$). Due to preferred orientation, selected peaks chosen at high and low 2θ values, these maps can solely be used for a qualitative interpretation.

(4) The resistive transitions shown in Refs. [2] and [3] are rather narrow, 4K-10K in Ref. [2], ~20K in Ref. [3]. It seems to me the authors are suggesting that because the samples in the DAC are expected to have compounds with much lower H content than LaH₁₀, and correspondingly lower T_c's or no T_c's, this should give rise to much broader resistive transitions than observed. It would be useful if the paper would expand on this point, if possible quantitatively, if not at least qualitatively.

The Reviewer is correct that in Refs. [2] and [3], Figures 3 and 4, respectively, show quite sharp and clean resistive transitions, which does suggest the presence of mainly one phase. However, if one looks at the Extended Data Fig. 2 of Ref. [3], one sees multiples “steps” and a very wide temperature range until the drop to the lowest resistivity value, potentially explained by the presence of multiple phases. Extended Data Fig. 5 of the same paper shows sharp resistivity transitions, however with T_c's of 70, 110, 215 and 250 K, depending on the sample. This again can be relate to multi-phase composition of the sample. In our manuscript, we identify La-H compounds that can provide an explanation for these resistivity curves, highlighting the complexity of these systems. A small discussion on this was added to the manuscript.

Report of Reviewer #3

The study on compressed La-H system has attracted great attention since the finding of near-room superconductivity in LaH₁₀ at a pressure of ~180 GPa that was actually stimulated by the theoretical prediction. The following experiments also found other phases or stoichiometries of La-H system. The further research on these remarkable systems remains required to clarify the detailed compositions and structures. In this manuscript, these authors carried out the single-crystal X-ray diffraction investigation on lanthanum hydrides at a pressure range of 50-180 GPa, in combination with density functional theory calculations. The found several stoichiometries of lanthanum hydrides under high pressure. Among these newly found phases, as I thought, the LaH₆ and LaH₉ are quite interesting, since both are predicted to be unstable in the previous studies. Moreover, the authors also found other interesting compounds like LaCH₂, which will help to if carbon participated in the reaction between the metals and hydrogen in the future experiment. These results are new and looks reasonable. Therefore, I could recommend the publication of this manuscript after considering below comments and suggestions.

We thank the Reviewer for his/her positive assessment of our work and for recommending it for publication after modifications.

1. In Figure S5, what temperature is used for these TDEP simulations? How many time step and supercell size in the TDEP simulations.

We thank the Reviewer for making us aware that we did not explicitly state the temperature used to perform the TDEP simulations. To be consistent with experiments, we set the sampling temperature to 300 K. We added this clarification in the manuscript.

Regarding the details of the sampling approach, we have followed the very complete account given in Ref. [75] of our manuscript, where sampling is performed self-consistently with steps using an increasing number of samples. The convergence with respect to the number of self-consistent steps is material dependent, and can be very fast with only 16 supercells in the final step for the more harmonic LaH₄, or more involved. In the current case, it was never necessary to go beyond 64 samples in the final step, even in the much more strongly anharmonic LaH₉.

As indicated, the chosen supercells were rectangular with at least 5 Å side length, resulting in supercell sizes up to 180 atoms. This is in line with earlier studies showing a rapid convergence of force constants with supercell size in LaH due to the short-range nature of the interatomic forces in metallic systems (see reference [46] of our manuscript).

2. For LaH₆, LaH₉ and LaH₁₀ systems, the authors could provide the uncertainty for hydrogen stoichiometry, while not for La₄H₂₃. Could authors provide a reason or make pertinent discussion in the manuscript on that.

All compounds which a stoichiometry that was written with a “+ δ ” suggests that both the experimental volume per La atom and theoretical calculations indicate a higher hydrogen content than expected from the compound’s structure-type based on its space group and lanthanum arrangement. As seen in Figure 2b), all compounds with the exception of LaH₃ and La₄H₂₃ have a difference between the experimental volume per La atom and the calculated volume per La atom (assuming the stoichiometry from the common structure-type) of more than one hydrogen atom. Thus, as we reasonably confident about the hydrogen content for the LaH₃ and La₄H₂₃, we did not add the “+ δ ”.

3. Could the authors do the density of states simulations for YH₆, YH₉, LaH₆ and LaH₉, which will help make discussions by comparing the estimated T_c of LaH₆ and LaH₉ with YH₆ and YH₉. As requested by the Reviewer, we calculated the electronic DOS for YH₆, YH₉, LaH₆ and LaH₉. As seen below, both LaH₆ and LaH₉ have a slightly higher H eDOS at the Fermi energy than their Y-H counterparts. This could help promote a higher T_c in the La-H compounds.

However, we want to stress that we have not been able to determine the position and quantity of the hydrogen atoms in our LaH_{6+ δ} and LaH_{9+ δ} compounds experimentally, here above assumed to have the LaH₆ and LaH₉ structures common for RE-H compounds. This, combined with the fact that there is a discrepancy between the experimentally and the theoretically determined unit cell volumes when assuming common RE-H structure types, we prefer not to include these eDOS in our manuscript.

4. In Fig. 2a, the authors may should discuss the equation of states for LaH₉ and LaH₁₀ that is quite flat comparing with LaH₄ and La₄H₂₃.

We assume the Reviewer is referring to the colored bands that correspond to the experimental equation of state. As written in the caption of Figure 2, these bands act as guides to the eye and have no real underlying physical meaning. Indeed, these suffer from two main issues: i) the EoS parameters were obtained from very few (often two) experimental datapoints and ii) the hydrogen content, for a given lanthanum atom arrangement, is thought to vary, which renders inaccurate the EoS parameters. This was clarified in the figure's caption.

5. The authors should carefully check all the references. For example, the page number is incorrect for Ref. 9, 43, and 70.

We thank for Reviewer for pointing this out. These three references were corrected and others were double-checked.

REVIEWER COMMENTS

Reviewer #1 (Remarks to the Author):

The authors have addressed my comments, and I am happy to recommend publication of this version. I suggest that the authors include their computed convex hull, and eDOS, with caveats in the SI.

Reviewer #2 (Remarks to the Author):

I thank the authors for their response to my review and the changes made, particularly the addition of Figs. S2 and S3 which I think provide important information. I wish the paper would have a clearer discussion of the implications of these results in connection with what are believed to be "superconducting" resistive transitions: the paper suggests that the determination of the T_c value could be problematic, but nowhere raises the possibility that the resistance drops observed may not be due to superconductivity. Anyway, this will surely be eventually clarified, and this paper is an important step in that direction.

Regarding Fig. S1: It took me a while to understand that the green color in (c) is superposition of blue and yellow, the caption says for (c) "blue regions" when there is no blue. That should be clarified. Also I don't understand how to understand that diamond powder and LaH_x compounds are in the same region of space, as seen in (d), (e). Are they intermixed, is one on top of the other, or what? The paper should say how the authors understand this.

I recommend publication after the authors have considered these points.

Reviewer #3 (Remarks to the Author):

The authors address all the issues raised by this reviewer. Now, I could recommend the publication of this manuscript.

Response to referees' comments:

To better structure our response to the referees' reports, each of their comments has been copied and our response to each of these comments is written in blue. When changes were made to the manuscript, the original statement from our paper is copied and followed by the updated text. The same changes are also highlighted in the marked up manuscript file.

Report of Reviewer #1

The authors have addressed my comments, and I am happy to recommend publication of this version. I suggest that the authors include their computed convex hull, and eDOS, with caveats in the SI.

We thank the Reviewer for recommending our manuscript to be published. We maintain that it would be highly misleading to provide in the Supplementary Information the computed convex hull as well as the eDOS of LaH₉ and LaH₁₀ since, as emphasized in the manuscript, these are mainly based on compounds that we did not observe. Indeed, aside from LaH₃ and La₄H₂₃, we do not know the hydrogen content in these La-H solids due to the strong unit cell volume mismatch between experiments and calculations.

Once further investigations will be done and reasonable candidates compounds, with a calculated unit cell volume matching that of experiments, are found, convex hull and eDOS calculations including these solids will only then provide reliable and meaningful physical insights.

Report of Reviewer #2

I thank the authors for their response to my review and the changes made, particularly the addition of Figs. S2 and S3 which I think provide important information. I wish the paper would have a clearer discussion of the implications of these results in connection with what are believed to be "superconducting" resistive transitions: the paper suggests that the determination of the T_c value could be problematic, but nowhere raises the possibility that the resistance drops observed may not be due to superconductivity. Anyway, this will surely be eventually clarified, and this paper is an important step in that direction.

We agree with the Reviewer that clarifying whether the resistive transitions are due to superconductivity or another mechanism is very important. However, as X-ray diffraction data—our main characterisation method—provides no information whatsoever on the transport properties of the formed La-H compounds, we feel that we could only speculate on the matter rather than providing the needed data-backed answers.

Regarding Fig. S1: It took me a while to understand that the green color in (c) is superposition of blue and yellow, the caption says for (c) "blue regions" when there is no blue. That should be clarified.

We agree with the Reviewer that the caption of Supplementary Figure 1 is not very clear on this matter. It has been modified to prevent any confusion.

Also I don't understand how to understand that diamond powder and LaH_x compounds are in the same region of space, as seen in (d), (e). Are they intermixed, is one on top of the other, or what? The paper should say how the authors understand this.

Indeed, the La-H compounds and diamond powder are definitely one on top of the other, and also likely to be partially intermixed. Essentially, sample regions initially with paraffin are expected, after laser-heating, to contain diamond powder. Paraffin was surrounding the La piece, especially on top of it but also in the irregularities of the La surfaces and on its sides and bottom, hence the presence of diamond powder essentially everywhere. This explanation was added to the caption of Supplementary Figure 1.

I recommend publication after the authors have considered these points.

We thank for Reviewer for their positive assessment of our work.

Report of Reviewer #3

The authors address all the issues raised by this reviewer. Now, I could recommend the publication of this manuscript.

We thank for Reviewer for their positive assessment of our work.